# DiffuLT: Diffusion for Long-tail Recognition Without External Knowledge

**Jie Shao**     **Ke Zhu**     **Hanxiao Zhang**     **Jianxin Wu**[*]
National Key Laboratory for Novel Software Technology, Nanjing University, China
School of Artificial Intelligence, Nanjing University, China
`{shaoj, zhuk, zhanghx}@lamda.nju.edu.cn, wujx2001@nju.edu.cn`

## Abstract

This paper introduces a novel pipeline for long-tail (LT) recognition that diverges from conventional strategies. Instead, it leverages the long-tailed dataset itself to generate a balanced proxy dataset without utilizing external data or model. We deploy a diffusion model trained from scratch on only the long-tailed dataset to create this proxy and verify the effectiveness of the data produced. Our analysis identifies approximately-in-distribution (AID) samples, which slightly deviate from the real data distribution and incorporate a blend of class information, as the crucial samples for enhancing the generative model's performance in long-tail classification. We promote the generation of AID samples during the training of a generative model by utilizing a feature extractor to guide the process and filter out detrimental samples during generation. Our approach, termed Diffusion model for Long-Tail recognition (DiffuLT), represents a pioneer application of generative models in long-tail recognition. DiffuLT achieves state-of-the-art results on CIFAR10-LT, CIFAR100-LT, and ImageNet-LT, surpassing leading competitors by significant margins. Comprehensive ablations enhance the interpretability of our pipeline. Notably, the entire generative process is conducted without relying on external data or pre-trained model weights, which leads to its generalizability to real-world long-tailed scenarios.

## 1 Introduction

Deep learning has exhibited remarkable success across a spectrum of computer vision tasks, especially in image classification, e.g., as exhibited by He et al. [2016], Dosovitskiy et al. [2021], Liu et al. [2021]. These models, however, encounter obstacles when faced with real-world long-tailed (LT) data, where the majority classes have abundant samples but the minority ones are sparsely represented. The intrinsic bias of deep learning architectures towards more populous classes exacerbates this issue, leading to sub-optimal recognition of minority classes despite their critical importance in practical applications.

Conventional long-tailed learning strategies such as re-weighting (Lin et al. [2017], Cao et al. [2019a]), re-sampling (Zhou et al. [2020a], Zhang et al. [2021a]), and structural adjustments (Wang et al. [2020], Cui et al. [2022]), share a commonality: they acknowledge the data's imbalance and focus on the training of models. They demand meticulous design and are challenging to generalize. Recently, a shift towards studying the dataset itself and involving more training samples through external knowledge to mitigate long-tailed challenges has emerged (Zhang et al. [2021b, 2024a]). Yet, in many real-world scenarios, access to specialized data is limited and closely guarded, such as in military or medical contexts. This predicament prompts a critical inquiry: *Is it feasible to balance long-tailed datasets **without** depending on external resources or models?*

---

[*]J. Wu is the corresponding author.

38th Conference on Neural Information Processing Systems (NeurIPS 2024).

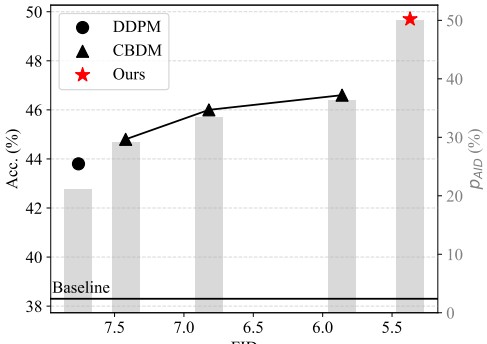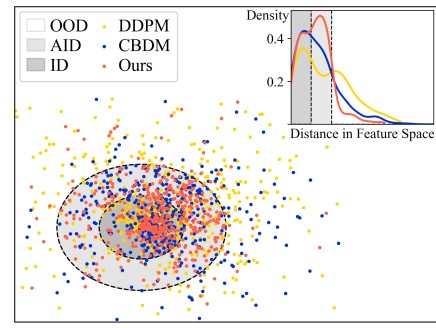

Figure 1: The samples generated by diffusion models improve long-tail classification on CIFAR100-LT, showing a correlation between FID and accuracy and a stronger correlation between the proportion of AID samples and accuracy. Our method significantly boosts classifier accuracy compared with others (left). Feature space visualization reveals that different diffusion models generate samples with varying distributions, and our model biases the generative process toward AID samples (right).

Our answer is *yes*. Recent advances in diffusion models have demonstrated their significant potential in generating high-quality images (Ho et al. [2020], Song et al. [2020], Rombach et al. [2022]). Assuming that diffusion models are proficient at learning distributions, we develop *a diffusion model trained from scratch* on *only* the long-tail distributed dataset. This model creates new samples for underrepresented classes, which are then used to train a classifier on the re-balanced dataset, leading to improved accuracy. We are the first to demonstrate the effectiveness of using generated samples, without relying on external data or models, in improving long-tail classification. We observe a notable pattern: enhancing the performance of the generative model with a loss modification called CBDM (Qin et al. [2023]) also enhances the classifier's accuracy, as illustrated in fig. 1. This phenomenon implies that a generative model with better performance tends to generate samples that are more beneficial for the classification task. This observation raises an important question: What are *the most valuable generated samples* for classification, and how are they generated? This question is critical, as it determines whether a diffusion model is going to be beneficial or detrimental for LT recognition.

We answer this question by analyzing features of generated samples, as visualized in fig. 1 using t-SNE (Van der Maaten and Hinton [2008]). We categorize the generated samples into three groups: in-distribution (ID), approximately in-distribution (AID), and out-of-distribution (OOD). Our research indicates that AID samples are pivotal in enhancing classifier performance. Through experiments, we conclude that a diffusion model can assimilate patterns from the head classes and integrates them into the tail ones to produce AID samples. These samples significantly enhance the quantity and diversity of the tail classes, thereby substantially improving their performance. Then the important question to solve is: How can we generate AID samples efficiently?

To encourage the model to predominantly generate AID samples, we introduce a novel type of loss. This loss employs a feature extractor to penalize the generation of ID and OOD samples. Such a strategy not only elevates the performance of the generative model on long-tail datasets but also renders it more effective and efficient in enhancing classifier performance.

In general, we introduce a new pipeline, DiffuLT (*Diffu*sion model for *Long-T*ail recognition), for long-tail datasets. It has three steps: initial training, sample generation, and retraining. Initially, we train a feature extractor and a diffusion model incorporating a supervision term to encourage the generation of AID samples. Subsequently, this generative model is employed to augment the dataset towards balance. The final step involves training a new classifier on the enriched dataset, with a minor adjustment to reduce the impact of synthetic samples. It is crucial to underscore the importance of training the diffusion model *without external data or knowledge, to maintain fairness in comparison*.

Our contributions are summarized as follows:

- We pioneer in addressing long-tail recognition by synthesizing images using diffusion models without relying on external data.

- Our research delves into the mechanisms underlying our approach, highlighting the significance of the generated AID samples. These samples emerge from a fusion of information from both head and tail classes, playing a crucial role in enhancing classifier performance.
- We introduce a novel loss function that enhances the performance of diffusion models on long-tailed datasets and biases them towards generating AID samples, thereby making the generation process more effective and efficient for classification.

Extensive experimental validation across CIFAR10-LT, CIFAR100-LT, and ImageNet-LT datasets demonstrates the superior performance of our method over existing approaches.

## 2 Related Work

**Long-tailed recognition** Long-tailed recognition is a challenging and practical task (Cui et al. [2019], Zhou et al. [2020b], Cao et al. [2019b], Zhang et al. [2023a], Zhu et al. [2024]), since natural data often constitute a squeezed and imbalanced distribution. The majority of traditional long-tailed learning methods can be viewed as (or special cases) of re-weighting (Cao et al. [2019b], Kang et al. [2020], Zhong et al. [2021a], Wang et al. [2024a]) and re-sampling (Cui et al. [2019]), with more emphasis on the deferred tail class to seek an optimization trade-off. There are variants of them that adopt self-supervised learning (Zhu et al. [2023], Li et al. [2021]), theoretical analysis (Li et al. [2022], Menon et al. [2021], Yang et al. [2024]) and decoupling pipeline (Kang et al. [2020], Zhou et al. [2020b]) to tackle long-tailed learning from various aspects, and they all achieve seemingly decent performance in downstream tasks.

One of the core difficulties in long-tailed learning is the *insufficiency of tail samples*. And recently, quite some works start to focus on this aspect by *involving more training samples through external knowledge* (Zhang et al. [2021b], Ramanathan et al. [2020], Dong et al. [2022], Shi et al. [2023a]). Nevertheless, the most distinct drawback of these works is that they either rely on *external data source* or *strong model weights*. This condition can seldomly hold true in practical scenarios where only a handful of *specialized* data are available and are secretly kept (consider some important military or medical data). We thus raise a natural question about long-tailed learning: *can we utilize the advantage of generating tail samples without resorting to any external data or model?* That is, the whole process is done in an in-domain (also called held-in) manner. In this paper, we propose to adopt the off-the-shelf diffusion model to learn and generate samples from the data at hand.

**Diffusion models and synthetic data** Diffusion models have been highly competitive in recent years (Ho et al. [2020], Song et al. [2020]), producing promising image quality in both unconditional and conditional settings (Dhariwal and Nichol [2021], Rombach et al. [2022], Ramesh et al. [2021]). Despite the predominant use in creating digital art, the application of diffusion models in scenarios of limited data remains under-explored. This paper affirms the utility of diffusion models in enhancing representation learning, particularly within the long-tailed learning framework, offering a novel insight into their application beyond conventional generative tasks.

The integration of synthetic data into deep learning, generated through methods like GANs Goodfellow et al. [2014], Isola et al. [2017] and diffusion models (Dhariwal and Nichol [2021], Rombach et al. [2022]), has been explored to enhance performance in image classification (Kong et al. [2019], Azizi et al. [2023], Zhang et al. [2024a], Trabucco et al. [2023]), object detection (Zhang et al. [2023b]), and semantic segmentation (Zhang et al. [2021c, 2023b]). These approaches often depend on substantial volumes of training data or leverage pre-trained models, such as Stable Diffusion, for high-quality data generation. Yet, the efficacy of generative models and synthetic data under the constraint of limited available data and in addressing imbalanced data distributions remains an unresolved inquiry. This paper specifically addresses this question, evaluating the viability of generative models and synthetic data in scenarios where data is scarce and imbalanced.

## 3 Method

### 3.1 Preliminaries

For image classification, we have a long-tail dataset $\mathcal{D} = \{(x_i, y_i)\}_{i=1}^{N}, y_i \in \mathcal{C}$ with each $x_i$ representing an input image and $y_i$ representing its corresponding label from the set of all classes

Table 1: FID of different generation models and their corresponding classifiers' accuracy.

| Model | FID | Acc. (%) |
|---|---|---|
| Baseline | - | 38.3 |
| DDPM | 7.76 | 43.8 |
| CBDM ($\tau = 3$) | 7.42 | 44.8 |
| CBDM ($\tau = 2$) | 6.82 | 46.0 |
| CBDM ($\tau = 1$) | 5.86 | 46.6 |

Table 2: Percentage of different types of generated samples for each model.

| Model | $p_{ID}$ | $p_{AID}$ | $p_{OOD}$ |
|---|---|---|---|
| DDPM | 39.1 | 21.2 | 39.7 |
| CBDM ($\tau = 3$) | 38.6 | 29.1 | 32.3 |
| CBDM ($\tau = 2$) | 40.2 | 33.5 | 26.3 |
| CBDM ($\tau = 1$) | 44.8 | 36.3 | 18.9 |

$\mathcal{C}$. In the long-tail setting, a few classes dominate with many samples, while most classes have very few images, leading to a significant class imbalance. The classes in $\mathcal{C}$ are ordered by sample count with $|c_1| \geq |c_2| \geq ... \geq |c_M|$, where $|c_j|$ denotes the number of training samples in class $c_j$ and $|c_1| \gg |c_M|$. The ratio $r = \frac{|c_1|}{|c_M|}$ is defined as the long-tail ratio. The goal of long-tail classification is to learn a classifier $f_\varphi : \mathcal{X} \to \mathcal{Y}$ capable of effectively handling the tail classes.

The naive idea is to train a generative model $\theta$ on the long-tail dataset $\mathcal{D}$ and use the trained model to generate new samples and supplement the tail classes. Inspired by its superior performance, we select diffusion models as the generative model in our pipeline. In our approach, we follow the Denoising Diffusion Probabilistic Model (DDPM by Ho et al. [2020]) framework. Given a dataset $\mathcal{D} = \{x_i, y_i\}_{i=1}^N$, we train a diffusion model to maximize the likelihood of the dataset. At every training step, we sample a mini-batch of images $\boldsymbol{x}_0$ from the dataset and add noise to obtain $\boldsymbol{x}_t$,

$$q(\boldsymbol{x}_t \mid \boldsymbol{x}_0) = \mathcal{N}(\sqrt{\bar{\alpha}_t}\boldsymbol{x}_0, (1 - \bar{\alpha}_t)\boldsymbol{I}), \tag{1}$$

where $\bar{\alpha}_t = \prod_{i=1}^t (1 - \beta_i)$ is calculated through pre-defined variance schedule $\{\beta_t \in (0,1)\}_{t=1}^T$. After training a diffusion model $\theta$ to get $p_\theta(\boldsymbol{x}_{t-1} \mid \boldsymbol{x}_t, t)$, we reverse the above process step by step to recover the original image $\boldsymbol{x}_0$ from pure noise $\boldsymbol{x}_T \sim \mathcal{N}(\boldsymbol{0}, \boldsymbol{I})$. The training objective is to reduce the gap between the added noise in forward process and the estimated noise in reverse process:

$$L_{\text{DDPM}} = \mathbb{E}_{t\sim[1,T],\boldsymbol{x}_0,\epsilon_t}[\|\boldsymbol{\epsilon}_t - \boldsymbol{\epsilon}_\theta(\sqrt{\bar{\alpha}_t}\boldsymbol{x}_0 + \sqrt{1 - \bar{\alpha}_t}\boldsymbol{\epsilon}_t, t)\|^2], \tag{2}$$

where $\epsilon_t \sim \mathcal{N}(\boldsymbol{0}, \boldsymbol{I})$ is the noise added to original images and $\boldsymbol{\epsilon}_\theta$ is the noise estimated by the trainable model with parameters $\theta$. DDPM can be conditional by transforming $y$ into a trainable class embedding and incorporating the label $y$ directly as am input, similar to time $t$. To improve the performance of DDPM on long-tailed dataset, several works (Qin et al. [2023], Zhang et al. [2024b]) have been proposed to adjust the distribution of generated samples. CBDM adds a distribution adjustment regularizer at the loss term. This term is designed to promote the generation of samples for tail classes, which is defined as (where sg means stop gradient):

$$L_{\text{CBDM}} = \frac{\tau t}{|\mathcal{Y}|} \sum_{y' \in \mathcal{Y}} (\|\boldsymbol{\epsilon}_\theta(\boldsymbol{x}_t, t, y) - \text{sg}(\boldsymbol{\epsilon}_\theta(\boldsymbol{x}_t, t, y'))\|^2 + \gamma \|\text{sg}(\boldsymbol{\epsilon}_\theta(\boldsymbol{x}_t, t, y)) - \boldsymbol{\epsilon}_\theta(\boldsymbol{x}_t, t, y')\|^2). \tag{3}$$

### 3.2 DiffuLT: Diffusion model for Long-Tail recognition

**Diffusion model helps long-tail classification.** In this phase, a *randomly initialized* diffusion model $\theta$ is trained to enrich the dataset. Preliminary experiments involve training a DDPM on a long-tailed dataset and using it to generate additional data. A threshold $N_t$ is set, and for classes $c_j$ with fewer than $N_t$ samples, we generate the images to meet this threshold. This augmentation results in a collection of synthetic samples, $\mathcal{D}_{\text{gen}} = \{(x_i, y_i)\}_{i=1}^{N_{\text{gen}}}$, where $N_{\text{gen}} = \sum_{c_j \in \mathcal{C}} \max(0, N_t - |c_j|)$ represents the total number of generated samples. These generated samples are then integrated with the original dataset, forming an augmented dataset $\mathcal{D} \cup \mathcal{D}_{\text{gen}}$, on which a classifier is trained to enhance classification performance.

We conducted experiments on CIFAR100-LT with an imbalance ratio of 100 and set $N_t = 500$ to supplement the data. The results, detailed in the second line of table 1, show a 5.5% accuracy increase for the classifier trained on $\mathcal{D} \cup \mathcal{D}_{\text{gen}}$ compared to the baseline. This improvement underscores the effectiveness of our straightforward method in boosting overall performance. Considering the generated samples (especially for tail classes) may be of lower quality due to limited data availability,

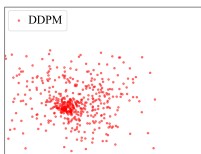 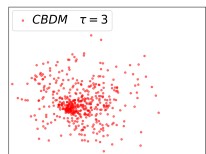 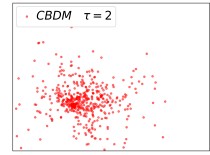 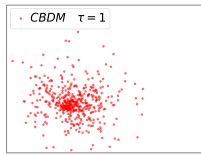

Figure 2: Visualization of generated samples for class 90 in feature space using t-SNE. The associated model is indicated in the upper-left corner.

Table 3: Quantities, overall classifier enhancement, and average improvement per sample for different groups of data generated by diffusion model.

| Group | $\|\mathcal{D}_{\text{gen}}\|$ | Acc. (%) | $\Delta\text{Acc}/\|\mathcal{D}_{\text{gen}}\|$ |
|---|---|---|---|
| Baseline | - | 38.3 | |
| ID | 21,511 | 44.2 | $2.75 \times 10^{-4}$ |
| AID | 11,886 | 45.2 | $5.78 \times 10^{-4}$ |
| OOD | 5,756 | 36.2 | $-3.61 \times 10^{-4}$ |

Table 4: Diffusion trained with varying proportions of head class data and the corresponding results for tail classes.

| $p_h$ | $p_{AID}$ | $\text{Acc}_t$ (%) |
|---|---|---|
| - | - | 25.0 |
| 0 | 25.8 | 26.0 |
| 40 | 33.2 | 29.7 |
| 80 | 35.7 | 32.5 |
| 100 | 39.1 | 32.8 |

Class-Balancing Diffusion Models (CBDM) is employed to improve generation quality in long-tailed settings. By integrating $L_{\text{DDPM}}$ and $L_{\text{CBDM}}$ in training the model $\theta$ on $\mathcal{D}$, the dataset is enhanced, and a classifier is trained as described previously. Subsequent testing on CIFAR100-LT reveals that the classifier achieves an accuracy of 46.6%, marking an 8.3% increase over the baseline, as noted in the final line of table 1.

**What samples are helpful? AID samples!** We adjusted the hyper-parameter $\tau$ in $L_{\text{CBDM}}$ and evaluate models with varying FID scores. Results presented in table 1 show that accuracy improves as FID decreases. Lower FID scores indicate that generated samples more closely resemble the real data distribution. Notably, some generated samples clearly fail, while others correctly resemble their intended class. This observation motivates further investigation into the efficacy of samples.

Class 90 (truck) is selected randomly as a representative example in CIFAR100-LT. A baseline classifier ($\varphi_0$), trained exclusively on the original dataset $\mathcal{D}$, is used to analyze the generated data. This classifier extracts features which are then visualized using t-SNE, as shown in fig. 2. The visualization reveals that samples generated via CBDM tend to be more centralized. For deeper analysis, we define the center $f_o$ of a class's features as the average of the real data in feature space, and set the maximum Euclidean distance between two real samples' features as a threshold $d_f$. We then define 3 types of the generated samples based on their distance to $f_o$:

$$d_i = \|f_i - f_o\|_2 : \quad \begin{cases} d_i \leq d_f, & \text{ID} \\ d_f < d_i \leq 2d_f, & \text{AID} \\ d_i > 2d_f, & \text{OOD} \end{cases} \quad (4)$$

where ID denotes in-distribution samples, which closely match the patterns of the original data. We define and name approximately in-distribution (AID) samples, which exhibit slight deviations. OOD stands for out-of-distribution ones, which are significantly differing from the center. We summarize the composition of samples generated by each model in table 2. Notably, the CBDM model generates a lower proportion of OOD samples, consistent with its FID score. For evaluating the impact of each type, we train classifiers using only the ID, AID, and OOD samples generated by CBDM with $\tau = 1$ respectively as $\mathcal{D}_{\text{gen}}$, combined with $\mathcal{D}$, and present the results in table 3. Surprisingly, classifiers trained with AID samples achieve the highest accuracy and show the greatest average improvement per sample. Based on this finding, our hypothesis is that *AID samples are the most beneficial in enhancing classifier performance*.

**Mechanisms behind the AID samples.** We conducted experiments to explore how AID samples enhance classifier performance and where their new and useful information originates. A diffusion model (CBDM with $\tau = 1$) is trained using images from tail classes (fewer than 100 samples), supplemented by a variable proportion $p_h$ of head class images. This model generates samples

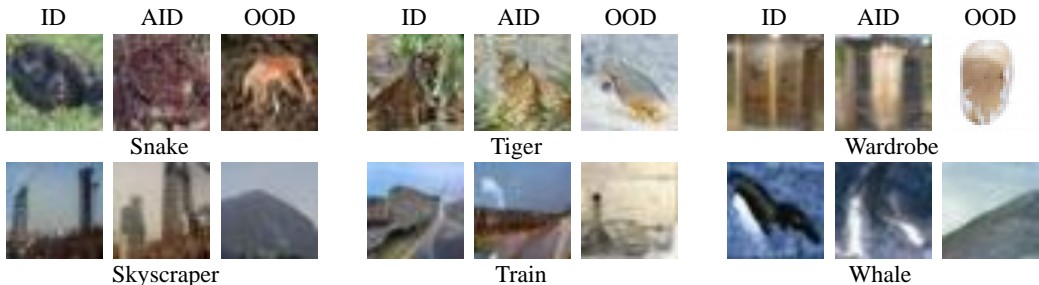

Figure 3: Examples of three groups of generated samples.

specifically for tail classes with the proportion of AID samples $p_{AID}$, and gets the performance of the corresponding classifier denoted as $\text{Acc}_t$. The results, presented in table 4, show that at $p_h = 0\%$, relying solely on tail class images, $p_{AID}$ is 25.8%, and $\text{Acc}_t$ improves marginally to 26.0%, only 1% above the baseline. As $p_h$ increases, both $p_{AID}$ and $\text{Acc}_t$ rise, peaking at $p_h = 100\%$. This trend illustrates the diffusion model's ability to transfer information from populous to underrepresented classes, effectively blending data across different classes into AID samples. Examples of the sample groups are displayed in fig. 3, where ID samples closely resemble real images, AID samples blend patterns from multiple classes, and OOD samples typically exhibit anomalies.

**Generation of AID samples**. How can we efficiently generate AID samples? While a filtering strategy can be used to collect AID samples, it is not the most efficient method. A more effective approach involves encouraging the generation model to specifically produce AID samples. Given that AID samples are defined by their distance from the center of real images in feature space, we can utilize the baseline classifier $\varphi_0$ as a feature extractor to guide the generation of AID samples. Our goal is to encourage a controlled deviation within feature space. After $T$ denoising steps, the deviation should ideally be within the range of $d_f$ to $2d_f$. Assuming that the deviation in each step is proportional to the noise strength, we introduce an additional term in the loss function to encourage small, stepwise deviations. We define the deviation at each step in feature space as

$$d_t = \frac{\sqrt{1 - \bar{\alpha}_T}}{\sqrt{1 - \bar{\alpha}_t}} \|\varphi_0(\boldsymbol{x}_0) - \varphi_0(\boldsymbol{x}_0 + \boldsymbol{\epsilon}_t - \boldsymbol{\epsilon}_\theta(\boldsymbol{x}_t, t, y))\|_2, \tag{5}$$

where $\varphi_0(\boldsymbol{x}_0 + \boldsymbol{\epsilon}_t - \boldsymbol{\epsilon}_\theta(\boldsymbol{x}_t, t, y))$ represents the de-noised images' feature. The new AID loss is then

$$L_{\text{AID}} = \alpha \mathbb{E}_{t \sim [1,T], \boldsymbol{x}_0, \epsilon_t} \|d_t - \frac{3}{2} d_f\|^2 . \tag{6}$$

where $\alpha$ is a hyper-parameter and defaulted to 0.1. We incorporate this term into both $L_{\text{DDPM}}$ and $L_{\text{CBDM}}$ to train the generation model. After training, we use this model to generate data. During the generation process, we employ $\varphi_0$ to filter out harmful OOD samples, resulting in $\mathcal{D}_{\text{gen}}$. We then train the classifier using the combined dataset $\mathcal{D} \cup \mathcal{D}_{\text{gen}}$. Recognizing that generated data are less crucial than real images, we introduce a weighting term to the cross-entropy loss to adjust the influence of the generated samples:

$$L_{\text{cls}} = -\sum_{(x,y,y_g) \in \mathcal{D} \cup \mathcal{D}_{\text{gen}}} (\omega y_g + (1 - y_g)) \log \frac{\exp(f_{\varphi,y}(x))}{\sum_{i=1}^{M} \exp(f_{\varphi,c_i}(x))}, \tag{7}$$

where $\omega$ controls the weight of generated samples and is set to 0.3 by default. $y_g$ is an additional label assigned to each image $x$, which distinguishes between generated and original samples. Specifically, $y_g = 1$ is used for generated samples, while $y_g = 0$ marks the original ones.

### 3.3 Overall Pipeline and Discussion

Now we are ready propose a new pipeline called DiffuLT to address long-tail recognition. The pipeline is shown in fig. 4 with four steps:

- **Training:** Initially, we train a feature extractor $\varphi_0$ and a conditional, AID-biased diffusion model $\theta$ using the original long-tailed dataset $\mathcal{D}$ alone.

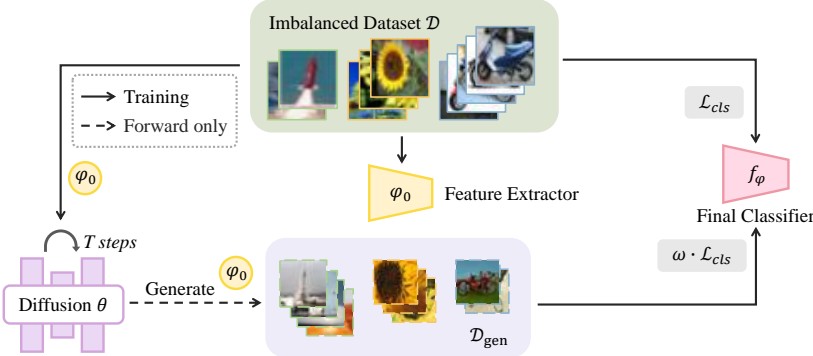

Figure 4: The overall pipeline of our method DiffuLT.

- **Generating:** We establish a threshold $N_t$ and employ the trained diffusion model $\theta$ to generate and supplement samples. Using $\varphi_0$, we filter out OOD samples, resulting in a refined dataset $\mathcal{D}_{\text{gen}}$.

- **Training:** We then train a new classifier $f_\varphi$ on the augmented dataset $\mathcal{D} \cup \mathcal{D}_{\text{gen}}$ using weighted cross-entropy, forming our final model.

Compared to traditional methods that focus primarily on training, ours not only enhances performance but is also reusable for model updates. Our method requires more training time, typically four times longer, to train the generation model and produce samples. However, our methods prove valuable when performance improvement is critical. Unlike typical data expansion methods, our approach offers both practical and theoretical benefits because it don't rely on any external dataset or model. For detailed analysis, please refer to the appendix B.

## 4 Experiment

### 4.1 Experimental setup

**Datasets.** Our research evaluate three long-tailed datasets: CIFAR10-LT (Cao et al. [2019a]), CIFAR100-LT (Cao et al. [2019a]), and ImageNet-LT (Liu et al. [2019a]). Following the methodology described in (Cao et al. [2019a]), we construct long-tailed versions of the first two datasets by adjusting the long-tail ratio $r$ to 100, 50, and 10 to test our method against various levels of imbalance.

**Baselines.** In our comparative analysis, we benchmark against a broad spectrum of classical and contemporary long-tailed learning strategies. The methods compared can be classified into multiple genres like re-weighting and re-sampling techniques, head-to-tail knowledge transfer approaches, data-augmentation, and so on. Some methods have issues such as unfair comparisons or implementation problems. We document both their results and our implementation outcomes in the appendix A.

**Implementation.** We set $\alpha = 0.1$, and $\omega = 0.3$. The generation thresholds $N_t$ for CIFAR10-LT and CIFAR100-LT were fixed at 5000 and 500, respectively. We employ ResNet-32 as the classifier backbone. For ImageNet-LT experiments, we set a generation threshold of $N_t = 300$. The classifiers were based on ResNet-10 and ResNet-50 architectures with $\omega = 0.5$.

More details about the experimental setup are available in the appendix A.

### 4.2 Experimental Results

**Generative Results.** We assess the efficacy of our specially designed loss function, detailed in table 5. This function improves the FID by reducing OOD samples, while also increasing the number of AID samples and the classifier's accuracy. Further experiments in table 6 highlight the necessity of our training loss. For benchmarking, we use a basic filtering strategy for CBDM, with all models generating samples to meet the threshold $N_t$ for each class. The terms "Kept" and "G-Num" denote

Table 5: FID of diffusion model, proportion of samples, and corresponding classifier accuracy

| Method | FID | $p_{ID}$ | $p_{AID}$ | $p_{OOD}$ | Acc. (%) |
|---|---|---|---|---|---|
| DDPM | 7.76 | 39.1 | 21.2 | 39.7 | 43.8 |
| CBDM | 5.86 | 44.8 | 36.3 | 18.9 | 46.6 |
| Ours | 5.37 | 40.7 | 50.1 | 9.2 | 49.7 |

Table 6: Methods and types of retained samples, pre-filtering counts, and classification accuracy.

| Method | Kept | G-Num | Acc. (%) |
|---|---|---|---|
| CBDM | All | 39,153 | 46.6 |
| CBDM | AID | 108,684 | 48.1 |
| CBDM | ID & AID | 48,414 | 47.1 |
| Ours | All | 39,153 | 49.7 |

Table 7: Results on CIFAR100-LT and CIFAR10-LT datasets. The imbalance ratio $r$ is set to 100, 50 and 10. The highest-performing results are in bold, with the second-best in underline. Additionally, we present the results for different groups (many, medium, and few) in CIFAR100-LT with $r = 100$.

| Method | CIFAR100-LT | | | CIFAR10-LT | | | Statistics | | |
|---|---|---|---|---|---|---|---|---|---|
| | 100 | 50 | 10 | 100 | 50 | 10 | Many | Med. | Few |
| CE | 38.3 | 43.9 | 55.7 | 70.4 | 74.8 | 86.4 | 65.2 | 37.1 | 9.1 |
| Focal Loss Lin et al. [2017] | 38.4 | 44.3 | 55.8 | 70.4 | 76.7 | 86.7 | 65.3 | 38.4 | 8.1 |
| LDAM-DRW Cao et al. [2019a] | 42.0 | 46.6 | 58.7 | 77.0 | 81.0 | 88.2 | 61.5 | 41.7 | 20.2 |
| cRT Kang et al. [2019] | 42.3 | 46.8 | 58.1 | 75.7 | 80.4 | 88.3 | 64.0 | 44.8 | 18.1 |
| BBN Zhou et al. [2020a] | 42.6 | 47.0 | 59.1 | 79.8 | 82.2 | 88.3 | - | - | - |
| RIDE (3 experts) Wang et al. [2020] | 48.0 | - | - | - | - | - | 68.1 | 49.2 | 23.9 |
| CAM-BS Zhang et al. [2021a] | 41.7 | 46.0 | - | 75.4 | 81.4 | - | - | - | - |
| MisLAS Zhong et al. [2021b] | 47.0 | 52.3 | 63.2 | 82.1 | 85.7 | 90.0 | - | - | - |
| DiVE He et al. [2021] | 45.4 | 51.1 | 62.0 | - | - | - | - | - | - |
| CMO Park et al. [2022] | 47.2 | 51.7 | 58.4 | - | - | - | **70.4** | 42.5 | 14.4 |
| SAM Rangwani et al. [2022] | 45.4 | - | - | 81.9 | - | - | 64.4 | 46.2 | 20.8 |
| CUDA Ahn et al. [2023] | 47.6 | 51.1 | 58.4 | - | - | - | 67.3 | 50.4 | 21.4 |
| CSA Shi et al. [2023b] | 46.6 | 51.9 | 62.6 | 82.5 | 86.0 | 90.8 | 64.3 | 49.7 | 18.2 |
| ADRW Wang et al. [2024b] | 46.4 | - | 61.9 | 83.6 | - | 90.3 | - | - | - |
| H2T Li et al. [2023] | 48.9 | 53.8 | - | - | - | - | - | - | - |
| DiffuLT | 51.5 | 56.3 | 63.8 | 84.7 | 86.9 | 90.7 | 69.0 | 51.6 | 29.7 |
| DiffuLT + BBN | _51.9_ | _56.7_ | _64.0_ | _85.0_ | _87.2_ | **90.9** | 69.5 | _51.9_ | _30.2_ |
| DiffuLT + RIDE (3 experts) | **52.4** | **56.9** | **64.2** | **85.3** | **87.3** | _90.9_ | _70.3_ | **52.1** | **30.7** |

the types of samples retained and the total number of samples generated before filtering, respectively. Our methods enhance the generation process's efficiency and achieve the highest accuracy.

**CIFAR100-LT & CIFAR10-LT.** We benchmark our approach against a range of methods on the CIFAR100-LT and CIFAR10-LT datasets, with results detailed in table 7. The results not shown in the original papers are indicated as "-" in the table. On CIFAR100-LT, our method surpasses competing models, achieving accuracy improvements of 13.2%, 12.4%, and 8.1% compared with the baseline for $r = 100$, 50, and 10, respectively. On CIFAR10-LT, our model also demonstrates strong competitiveness, enhancing accuracy by 14.3%, 12.1%, and 4.3% across the long-tail ratios, further validating the effectiveness of our method. Since our methods solely modify the training data, they can be easily integrated with other methods to achieve better results.

For CIFAR100-LT with an imbalanced ratio of 100, performance is also assessed across three categories: many (classes with over 100 samples), medium (classes with 20 to 100 samples), and few (classes with fewer than 20 samples). While our approach does not lead in the "Many" category, it excels in "Med." and "Few", significantly outperforming others in the "Few" group with a 29.7% accuracy — 8.3% above the nearest competitor and 20.6% beyond the baseline.

**ImageNet-LT.** On the ImageNet-LT dataset, our methodology is evaluated against existing approaches, with results summarized in table 8. Utilizing a ResNet-10 backbone, our method registers a 50.4% accuracy, outperforming the nearest competitor by 4.5%. With ResNet-50, the accuracy further escalates to 56.4%, marking a substantial 14.8% enhancement over the baseline. Despite a slight decline in the "Many" category relative to the baseline, our approach excels in "Med." and 'Few", with the latter witnessing a remarkable 33.6% improvement over the baseline. Our method can be combined with others to achieve enhanced results.

Table 8: Results on ImageNet-LT. We deploy ResNet-10 and ResNet-50 as classifier backbones. Top-performing results are highlighted in bold, with second-best outcomes underlined.

| | ResNet-10 | ResNet-50 | | | |
|---|---|---|---|---|---|
| | All | All | Many | Med. | Few |
| CE | 34.8 | 41.6 | 64.0 | 33.8 | 5.8 |
| Focal Loss Lin et al. [2017] | 30.5 | - | - | - | - |
| OLTR Liu et al. [2019b] | 35.6 | - | - | - | - |
| cRT Kang et al. [2019] | 41.8 | 47.3 | 58.8 | 44.0 | 26.1 |
| RIDE (3 experts) Wang et al. [2020] | 45.9 | 54.9 | 66.2 | 51.7 | 34.9 |
| MisLAS Zhong et al. [2021b] | - | 52.7 | - | - | - |
| CMO Park et al. [2022] | - | 49.1 | **67.0** | 42.3 | 20.5 |
| SAM Rangwani et al. [2022] | | 53.1 | 62.0 | 52.1 | 34.8 |
| CUDA Ahn et al. [2023] | - | 51.4 | 63.1 | 48.0 | 31.1 |
| CSA Shi et al. [2023b] | 42.7 | 49.1 | 62.5 | 46.6 | 24.1 |
| ADRW Wang et al. [2024b] | - | 54.1 | 62.9 | 52.6 | 37.1 |
| DiffuLT | 50.4 | 56.4 | 63.3 | 55.6 | 39.4 |
| DiffuLT + RIDE (3 experts) | **51.1** | **56.9** | 64.1 | **55.8** | **39.9** |

Table 9: Ablation experiments to verify the effect of each module.

| Gen. | $L_{\mathrm{AID}}$ | Filt. | Weight | Acc. (%) |
|---|---|---|---|---|
| | | | | 38.3 |
| ✓ | | | | 46.6 |
| ✓ | ✓ | | | 49.7 |
| ✓ | ✓ | ✓ | | 50.3 |
| ✓ | ✓ | ✓ | ✓ | **51.5** |

Table 10: Performance with different weights $\omega$ and hyper-parameter $\alpha$.

| $\omega$ | Acc. (%) | $\alpha$ | Acc. (%) |
|---|---|---|---|
| 0 | 38.3 | 0 | 38.3 |
| 0.1 | 49.2 | 0.1 | 49.7 |
| 0.3 | 51.5 | 0.5 | 49.5 |
| 0.5 | 50.1 | 1.0 | 48.3 |
| 0.7 | 50.3 | 2.0 | 45.1 |
| 1.0 | 50.3 | 4.0 | 43.3 |

## 4.3 Ablation Study

**Different modules in our pipeline.** Our methodology comprises several critical components: generated samples (using CBDM), AID-biased loss, filtering, and weighted cross-entropy. We conduct ablation experiments on CIFAR100-LT with $r = 100$. The results, presented in table 9, highlight the crucial role each component plays in enhancing the overall performance. Notably, the generated samples and AID-biased loss are the most influential factors.

**Hyper-parameters.** We adjust the parameters $\omega$ in the weighted cross-entropy and $\alpha$ in $L_{\mathrm{AID}}$ on CIFAR100-LT with $r = 100$ and evaluate the classification results. These results are summarized in table 10. Through iterative adjustments, we find that the optimal performance, a 51.5% classification accuracy, is achieved when $\omega = 0.3$. Similarly, the best setting for $\alpha$ is determined to be 0.1. Consequently, we establish $\omega = 0.3$ and $\alpha = 0.1$ as the default settings for our method.

## 5 Conclusion

In this research, we proposed a novel, data-centric approach designed to address the challenges of long-tail classification. We defined and identified AID (approximately in-distribution) samples as the important ones. We then revised a diffusion model trained with an AID-biased loss term on only the original dataset for the purpose of generating more AID samples, thereby significantly enriching the dataset. Following sample generation, we trained a classifier on this enhanced dataset and employed a weighted cross-entropy loss. Our method has shown to deliver competitive performance, highlighting its efficacy in real-world applications. The experiments conducted as part of this study notably emphasize the critical role played by AID samples and their significant impact.

We propose that this approach introduces a new paradigm for tackling long-tail classification challenges, offering a substantial complement to existing methodologies. It provides a robust framework

that can be adapted to various scenarios where performance is a critical factor. Despite its advantages, the training of the diffusion model and the generation of samples are time-consuming. The need for optimization in training and generation speeds represents a limitation of our current method. We will leave this point as future work to further improve the effectiveness and efficiency of our method.

## Acknowledgements

We acknowledge the funding provided by the National Natural Science Foundation of China under Grant 62276123 and Grant 61921006. J. Wu is the corresponding author.

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

# A Experiment

## A.1 Experimental settings

Due to space constraints in the main paper, we only include essential information about the experimental setup. Additional details are provided here.

**CIFAR100-LT & CIFAR10-LT.** The original CIFAR100 and CIFAR10 datasets each consist of a training set with 50,000 images evenly distributed across 100 or 10 classes, respectively. The CIFAR100-LT and CIFAR10-LT datasets, derived from CIFAR100 Krizhevsky [2009] and CIFAR10 Krizhevsky [2009], feature a long-tail distribution where the class frequency decreases exponentially from class 0 to the last class. Commonly used long-tail ratios are 100, 50, and 10. Specifically, the CIFAR100-LT subsets contain 10,847, 12,608, and 19,573 images, with the largest class containing 500 samples and the smallest having 5, 10, and 50 samples, respectively. Similarly, the CIFAR10-LT subsets consist of 12,406, 13,996, and 20,431 images, with the largest classes containing 5,000 samples and the smallest 50, 100, and 500 samples, respectively.

Experiments on CIFAR100-LT and CIFAR10-LT utilize the framework and training methodologies from Qin et al. [2023], incorporating the CBDM Qin et al. [2023] loss function and Adaptive Augmentation Karras et al. [2020]. We set the training duration to 500,000 steps, with hyperparameters $\tau$ and $\gamma$ fixed at 1 and 0.25, as per the cited study. The batch size is maintained at 128, the diffusion process runs for 1,000 time steps, and the learning rate is set at 0.0002 using an Adam optimizer.

For classifier training, we follow the code and protocols from Zhou et al. [2020a], which prescribe a 200-epoch training regimen. The classifier training also employs a batch size of 128, utilizing an SGD optimizer with a learning rate of 0.1. The feature extractor $\varphi_0$ is trained using this setup without any additional methods or data. The final classifier is trained similarly but incorporates both generated and original samples. All training tasks are conducted on $8\times$ NVIDIA GeForce RTX 3090 GPUs, with further discussed in appendix B

**ImageNet-LT.** ImageNet-LT, comprising 115,846 images across 1,000 classes with a maximum of 1,280 images per class and a minimum of 5, follows the specifications set in (Liu et al. [2019a]). This dataset is derived from ImageNet Russakovsky et al. [2015] by sampling a subset according to a Pareto distribution with a power value of $\alpha = 6$. For perspective, the original ImageNet training set contains 1,281,167 images, making ImageNet-LT less than 10% the size of the original dataset. The test set of ImageNet-LT mirrors that of ImageNet, containing 100,000 images.

Due to its considerable size and image resolution, ImageNet-LT necessitates modifications from the standard CBDM framework to address inefficiencies. We adapt the codebase from Dhariwal and Nichol [2021] for this purpose, extending the training to 1,980,000 iterations. This setup uses the AdamW optimizer with a learning rate of 3e-4 and a batch size of 64, adjusted from the original 256 due to GPU memory constraints. Moreover, the model is trained at an image resolution of 256, and the total diffusion time step is set to 1,000.

Classifier training for ImageNet-LT employs the framework from Zhang et al. [2021a]. Specifically, the model is trained over 100 epochs with a batch size of 512 using the SGD optimizer at a learning rate of 0.2. All training tasks are carried out on $8\times$ NVIDIA GeForce RTX 3090 GPUs.

## A.2 Generated Images

We synthesize images for CIFAR-100, CIFAR-10, and ImageNet-LT to demonstrate their utility in enhancing long-tail recognition. Randomly selected examples of the generated samples are displayed, particularly for CIFAR100-LT in fig. 5, where images have a resolution of $32 \times 32$ and focus on "few-shot" classes—those with fewer than 20 instances. Despite the limited examples available in these classes, our diffusion model effectively utilizes the entire dataset to produce high-quality samples. While these synthesized images maintain some similarities with their original counterparts, the notable variations make them valuable for model training. However, because the generated images may sometimes distort essential features or introduce inaccuracies, it becomes crucial to filter out the OOD samples to maintain their usefulness.

In fig. 6, we showcase the generated images for CIFAR10-LT, which demonstrate superior quality compared to those from CIFAR100-LT, due to a more abundant training dataset. In CIFAR10-LT,

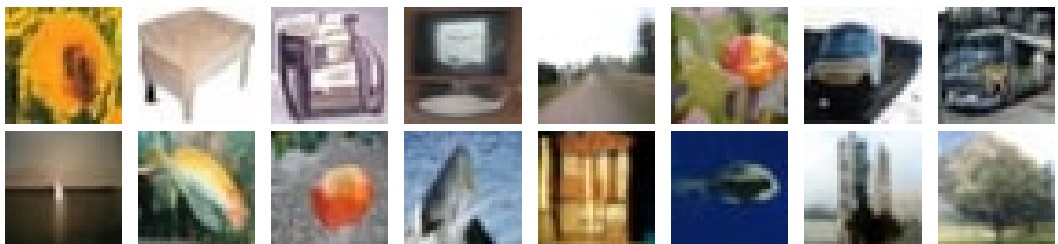

Figure 5: Generated images for CIFAR100-LT

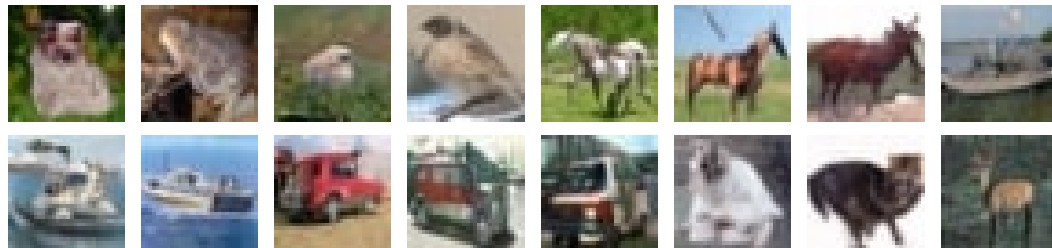

Figure 6: Generated images for CIFAR10-LT

Table 11: Repeated experiments on CIFAR100-LT and CIFAR10-LT to test the robustness of our methods.

| Method | CIFAR100-LT | | | CIFAR10-LT | | |
|---|---|---|---|---|---|---|
| | 100 | 50 | 10 | 100 | 50 | 10 |
| DiffuLT[1] | 51.5 | 56.3 | 63.8 | 84.7 | 86.9 | 90.7 |
| DiffuLT[2] | 51.7 | 56.3 | 63.7 | 84.9 | 86.8 | 90.7 |
| DiffuLT[3] | 51.5 | 56.3 | 63.3 | 84.9 | 86.3 | 90.6 |

Table 12: Repeated experiments on ImageNet-LT to test the robustness of methods.

| | ResNet-10 | ResNet-50 |
|---|---|---|
| DiffuLT[1] | 50.4 | 56.4 |
| DiffuLT[2] | 50.4 | 56.5 |
| DiffuLT[3] | 50.5 | 56.5 |

each class generally contains ten times more images than in CIFAR100-LT. The smallest class in CIFAR10-LT has 50 images, greatly exceeding the minimum of 5 in CIFAR100-LT. This increase in sample size, however, requires a more stringent filtering process to prevent potential information loss due to the larger volume of images generated.

For ImageNet-LT, the generated images, displayed in fig. 7, feature a resolution of $224 \times 224$, which is significantly clearer than those from CIFAR-10 and CIFAR-100. While some finer details, such as text within the image or textures like fur, may not be fully distinct, the generated images effectively capture the essential patterns of the classes. These images can significantly aid the long-tail recognition task, particularly for classes with fewer original samples.

### A.3 Robustness anlysis

Since CIFAR100-LT and CIFAR10-LT are typically sampled randomly from their original datasets, we tested our methods across various sampled sets to assess their effectiveness and robustness. The results, displayed in table 11, demonstrate that our method's performance is stable, exhibiting only minimal variations. For ImageNet-LT, which is a fixed dataset, we generated samples three times to

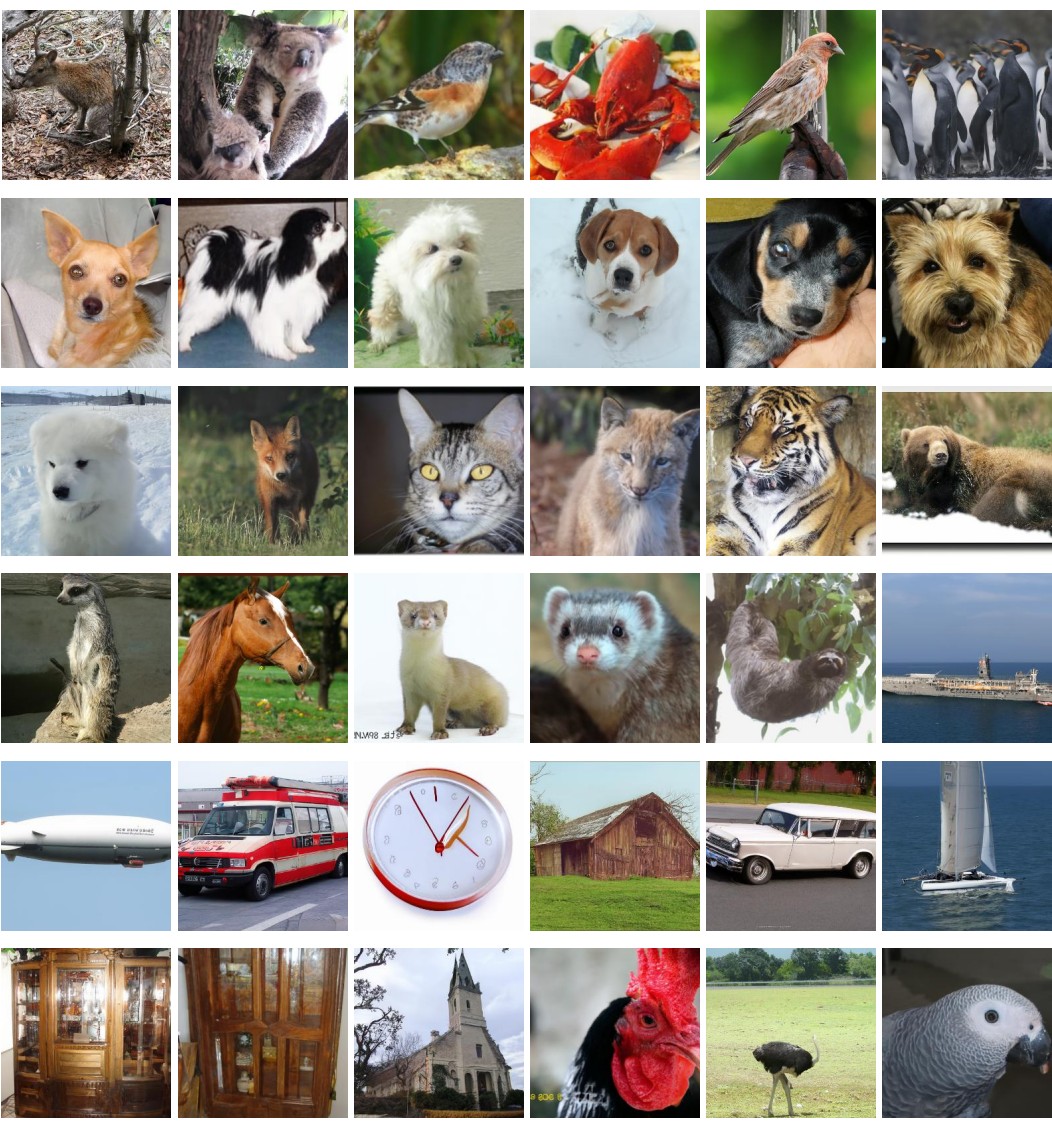

Figure 7: Generated images for ImageNet-LT

examine consistency. The variations among different sample sets are minimal, as shown in table 12. Therefore, our method proves to be robust in enhancing long-tail classification performance.

### A.4    Details of Baseline Methods.

In the main paper, we outline various approaches to long-tail classification and benchmark our methods against these strategies. Here, we provide a concise overview of the methods within each category. For re-weighting and re-sampling techniques, we examine Cross-Entropy (CE), Focal Loss (Lin et al. [2017]), LDAM-DRW (Cao et al. [2019a]), cRT (Kang et al. [2019]), BBN (Zhou et al. [2020a]), CSA (Shi et al. [2023b]), and ADRW (Wang et al. [2024b]). In the realm of head-to-tail knowledge transfer, we include methods such as OLTR (Liu et al. [2019b]) and H2T (Li et al. [2023]). Label-smoothing strategies are represented by MisLAS (Zhong et al. [2021b]) and DiVE (He et al. [2021]), while in data augmentation, we compare our approach with CAM-BS (Zhang et al. [2021a]), CMO (Park et al. [2022]), and CUDA (Ahn et al. [2023]). Lastly, SAM (Rangwani et al. [2022]) exemplifies an advanced optimization technique, and RIDE (Wang et al. [2020]) showcases a mixture of expert technique in our comparison.

Table 13: Results on CIFAR100-LT using an alternative pipeline based on the implementation guidelines from BSCE Ren et al. [2020], with imbalance ratios $r$ set at 100, 50, and 10.

| Method | CIFAR100-LT | | |
| --- | --- | --- | --- |
| | 100 | 50 | 10 |
| Baseline | 38.3 | 43.9 | 55.7 |
| Baseline* | 45.3 | 50.3 | 61.9 |
| BSCE Ren et al. [2020] | 50.8 | - | 63.0 |
| PaCo Cui et al. [2021] | 52.0 | 56.0 | 64.2 |
| GPaCo Cui et al. [2023] | 52.3 | 56.4 | 65.4 |
| DiffuLT | 54.7 | 58.9 | 66.1 |
| DiffuLT + GPaCo | 55.4 | 59.5 | 66.4 |

## A.5  Other Methods.

For various reasons, some methods are not included in our experimental comparisons. This decision primarily stems from two factors. Firstly, several methods, despite demonstrating impressive results, do not have publicly available code (Zada et al. [2022]), limiting our ability to perform direct comparisons. Secondly, methods such as those in Cui et al. [2021] and Cui et al. [2023] achieve commendable results and can be replicated. However, their comparisons may be considered unfair. These methods utilize AutoAugment (Cubuk et al. [2018]), with parameters optimized across the entire dataset, rather than specifically for the long-tailed segment. This approach significantly boosts their baseline performance, as reported in Ren et al. [2020]. For instance, on CIFAR100-LT with an imbalance ratio of 0.1, baseline accuracy improves from 38.3 to 45.3, as shown in table 13. The first "Baseline" line reflects the standard settings, while entries marked with * use the enhanced settings, demonstrating a substantial improvement. Comparing these results with those obtained under standard conditions would be unfair. Our methodology could also be adapted to such settings and combined with these techniques to achieve excellent results, as illustrated in table 13. However, these results are omitted from the main paper to maintain a fair comparison.

## B  Discussion

### B.1  Comparison of our method with other type of methods.

The comparison is divided into two parts: comparing our methods with traditional long-tail recognition approaches and contrasting them with data synthesis methods, as shown in fig. 8.

Compared to traditional long-tail classification methods that predominantly focus on training, our approach offers a novel perspective. Rather than designing intricate methods to facilitate training on long-tailed datasets, we straightforwardly enhance the dataset using a generative model. This approach is not only innovative but also compatible with existing training methodologies and demonstrates improved performance. However, the main limitation is the training of the generative model, which is time-consuming and challenging to scale. These points will be further discussed in the subsequent subsection.

When comparing with data synthesis methods, it is clear that our approach does not surpass those employing technologies like Stable Diffusion or CLIP on long-tailed datasets. Nevertheless, the use of such large models trained on extensive data eliminates the core challenge of long-tail problems—data scarcity. For instance, the class "train" in CIFAR100-LT has only ten images, whereas Stable Diffusion has been trained on thousands of train images. In practical settings, accessing such expansive models tailored to specific datasets is unrealistic. These methods mainly benefit from data leakage. Our approach, in contrast, is designed for real long-tail scenarios without reliance on external data or models, providing practical and theoretical value. We address several previously unanswered questions:

- Is a diffusion model trained from scratch beneficial for long-tail recognition? Yes.

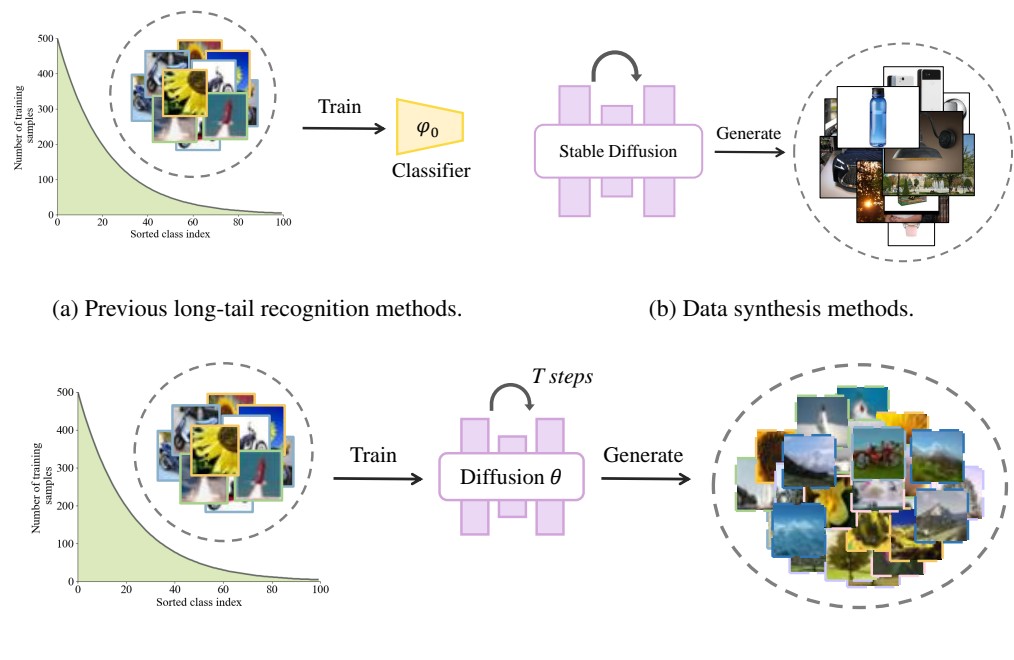

(a) Previous long-tail recognition methods.

(b) Data synthesis methods.

(c) Ours

Figure 8: Main caption describing all images

- Which generated samples are useful for long-tail recognition? AID samples.
- Why does diffusion work for long-tail recognition? It blends class information to generate beneficial and novel samples.

## B.2 Limitation

The primary limitation of our methods is the extensive training time required for the generative model. For instance, training a diffusion model on CIFAR100-LT takes 24 hours, while ImageNet-LT requires approximately six days. As the quality and quantity of data increase, the training costs scale up significantly, making it challenging to apply our methods to larger datasets such as iNaturalist and Places-LT due to resource and time constraints.

Despite these challenges, our methods are highly effective in addressing the long-tail recognition problem. To improve training efficiency, we are exploring two potential solutions. The first involves adopting techniques that accelerate the training and inference processes of diffusion models. The second strategy considers the use of pre-trained generative models in real long-tail scenarios. This approach does not contradict our previous assertion that using large, pre-trained models on familiar long-tailed data is unfair and not truly representative of long-tail challenges. Instead, we advocate for the use of pre-trained models on long-tail datasets they have not previously encountered, ensuring fairness and practical applicability. Employing a pre-trained model could significantly expedite our pipeline by eliminating the need to train from scratch. This topic extends beyond the scope of this paper and will be addressed in future research.

